# Supply Chain Management: A Review and Bibliometric Analysis

**Hui Fang** [1,2] , **Fei Fang** [3,*] , **Qiang Hu** [3,*] and **Yuehua Wan** [1,2]

1   Library, Zhejiang University of Technology, Hangzhou 310014, China
2   Institute of Information Resource, Zhejiang University of Technology, Hangzhou 310014, China
3   Zhejiang College, Shanghai University of Finance and Economics, Jinhua 321013, China
*   Correspondence: fangfei@shufe-zj.edu.cn (F.F.); huqiang@shufe-zj.edu.cn (Q.H.);
    Tel.: +86-138-5797-1486 (F.F.); +86-186-6713-7656 (Q.H.)

**Abstract:** Supply chain management (SCM), which generally refers to horizontal integration management, has steadily become the core competitiveness of company rivalry and an essential approach to developing national comprehensive and national strength since the end of the 20th century due to the numerous needs arising from a competitive international economy. Manufacturers develop a community of interest by forming long-term strategic partnerships with suppliers and vendors throughout the supply chain. This paper defines supply chain management by reviewing the existing literature and discusses the current state of supply chain management research, as well as prospective research directions. Specifically, we conducted a bibliometric analysis of the influential studies of SCM in terms of various aspects, such as research areas, journals, countries/regions, institutions, authors and corresponding authors, most cited publications, and author keywords, based on the 8998 reviews and articles collected from the SCI and SSCI database of the Web of Science (WoS) between 2010 and 2020. The results show that the major research areas were Management (3071, 34.13%), Operations Research & Management Science (2680, 29.78%), and Engineering, Industrial (1854, 20.60%) with TP and TPR%. The most productive journal and institution were J. Clean Prod and Hong Kong Polytech Univ with a TP of 554 and 238, respectively. China, USA, and UK were the top three contributing countries. Furthermore, "sustainability", "green supply chain (management)", and "sustainable supply chain (management)" were the most popular author keywords in recent three years and since 2010, apart from the author keywords of SCM. When combined with the most cited articles in recent years, the application of block chain and Industry 4.0 in supply chain management increased rapidly and generated great attention.

**Keywords:** supply chain management (SCM); bibliometric analysis; sustainability; block chain; industry 4.0

## 1. Introduction

Supply chain management (SCM) has become a worldwide hot spot in the research and practical application of enterprise management since the 1990s. Stevens [1] proposed that the supply chain was a connected series of activities concerned with planning, coordinating, and controlling material, parts, and finished goods from the supplier to customer. Lee and Billington [2] defined the supply chain as a network tool for enterprises to obtain raw materials to produce semi-finished or final products and deliver them to consumers through sales channels. They also believed that a supply chain was a network of facilities and distribution options that performed the functions of procurement of materials, transformation of these materials into intermediate and finished products, and distribution of these finished products to customers [3]. The concept of supply chain management first appeared in the 1980s, and a large number of articles emerged in the early 1990s. The Supply Chain Council of America defined the SCM as "encompassing every effort involved in producing and delivering a final product, from the supplier's supplier to the customer's customer". Copacino [4] argued that SCM is the art of managing the flow of materials

and products from source to user. Evas et al. [5] regarded SCM as a management model that connected suppliers, manufacturers, distributors, retailers, and end-users through feed-forward information flow, feedback material flow, and information flow. Balsmeier [6] considered SCM as a new management strategy that integrates different enterprises to increase the efficiency of the entire supply chain and pays attention to cooperation between enterprises, which differed from supplier management. Mentzer et al. [7] defined SCM as the systemic, strategic coordination of the traditional business functions and the tactics across these business functions within a particular company and across businesses within the supply chain, for the purposes of improving the long-term performance of the individual companies and the supply chain as a whole. According to these definitions, SCM has the effect of lowering costs and increasing customer value and satisfaction to compete successfully [8–10]. Understanding where supply chain management boundaries exist is beneficial for improving supply chain performance [11]. Croxton et al. [12] recognized the benefits of a process approach to managing the business and the supply chain.

To summarize this literature review, we define supply chain management as a holistic functional network chain, centering on the core enterprise, starting from the procurement of raw materials to the completion of the final product, and then through the enterprise's sales system and transportation network to deliver the finished product to each consumer on time, involving material suppliers, manufacturers, distributors, retailers until the end user. Compared with the traditional management model, SCM is holistic and focuses on strategic cooperation management. SCM can also optimize and integrate advantageous resources among the member enterprises in the supply chain, making full use of the internal and external resources of each enterprise and enhance the overall competitiveness. Moreover, the goal of SCM is not only to ensure the realization and completion of various market development but also to provide high-quality services for end customers and make them satisfied. In conclusion, SCM combines global strategy management with high flexibility and quick market reaction in a complex and dynamic competitive environment, as opposed to vertical integration. Because of the integration and administration of numerous organizations, SCM is complicated and dynamic. The limitations of SCM mainly exist in the process of the application practice. For example, both the interest conflicts with suppliers and the obvious competition among enterprises make it difficult to manage SCM. Several problems have been revealed in the development of SCM research and have resulted in limiting the effectiveness and efficiency of the supply chain applied in the enterprises. The following are the main problems.

(1)	SCM has become more complicated due to supply chain disruptions related to the risks of the supply chain. Many supply chains tend to break down and need a long time to recover when major disruptions occur. Scholars have defined different types of risks in the supply chain and emphasized the importance of supply chain risk management (SCRM) [13–17]. Among them, environmental variables have been studied for a number of reasons and were thought to influence the selection of appropriate organizational structures [18]. Simangunsong and Hendry [19] defined supply chain uncertainty as a problem that every practicing manager faced, and built a theoretical framework for future research by taking a broad perspective of supply chain uncertainty, which included supply chain risk. Typical disruptions of SCM include environmental uncertainty, sudden events, demand fluctuations, and other reasons. It was considered that the environmental uncertainty framework remained conceptual [20], and environmental uncertainty was associated with supply chain performance [21]. On the other hand, almost every industrial sector's demands seem to be more erratic than before, due to rising market turbulence. Christopher and Lee [22] considered external events, such as war, strikes, and terrorist attacks, as factors of market turbulence and uncertainty. The most significant sudden event in recent years has been the coronavirus (COVID-19), which has impacted practically every sector. COVID-19's effects on the supply chain have already attracted academics' concerns [23–25]. The COVID-19 outbreak illustrated how pandemics and epidemics

may severely disrupt global supply chains, emphasizing the necessity for flexibility in order to manage epidemic and demand risks [26]. Scholars also emphasized the importance of dealing with demand fluctuation disruptions that might disrupt the supply chain [27–29], in order continue to function supply chain smoothly.

(2)   Information-technology applications in SCM are still in their immaturity. Many scholars have emphasized the use of information technology (IT) in supply chain management (SCM) [30,31]. Zhong et al. [32] reviewed storage technology, data processing technology, data visualization techniques, big data analytics, and models and algorithms as the main current technologies. Ivanov et al. [33] studied the relationships between digitalization and SC disruption risks. The application value of radio-frequency identification (RFID) in supply chain management (SCM) had been discussed [34–37]. However, the great majority of existing theoretical models are based on comprehensive knowledge exchange and unrestricted information flow, while the fact is that information system operation is inefficient due to the technological barrier.

(3)   The theory of SCM has limits that cannot connect with practical industrial operations closely. Supply chain management has attracted considerable attention from the international academic and commercial sectors as one of the most important management theories and methodologies for enhancing organization competitiveness in the 1990s [38]. The relevance of integrating a company's supply chain strategy to its entire business plan has been discussed, as well as some practical supply chain management suggestions [39]. Sangari et al. [40] created a hybrid assessment approach that combined fuzzy logic, DEMATEL (decision-making trial and evaluation laboratory), and ANP (analytic network process) and applied it to an automobile firm that wanted to increase supply chain agility. However, the divergence between theoretical research and practical operations does exist [41,42]. Sweeney et al. [43] mentioned some key success factors and barriers to implementing SCM theory in practice, as well as some practical measures that can be implemented at the policy/supply chain/corporate level to increase the level of effective SCM adoption.

We used bibliometric analysis, which is a common and thorough approach for discovering and evaluating vast amounts of scientific data, as the research approach to analyze supply chain management (SCM) in this paper. Donthu et al. [44] believed that bibliometric analysis enabled us to examine the evolutionary subtleties of a specific discipline, while also providing insight into new areas. The limitations of bibliometric analysis were also mentioned by these authors. Bibliometric analysis has been used in a variety of review areas, such as different disciplines, industries, decision-making techniques, and smart technologies. As a result, bibliometric analysis has been used in management reviews, economics reviews [45], financial literacy reviews [46], and education reviews [47]. Scientific publications [48], artificial intelligence [49], and grey system theory [50] also use bibliometric analysis for review. Bibliometric analysis has been used as a reliable approach to identify hotspots and research trends in a variety of research fields, and we have also used this method to publish a number of articles in the fields of public health [51], medicine [52,53], mechanics [54], and social science [55,56]. However, limitations of this approach should also be noted. The h-index was created as a straightforward indicator of output and effect combined due to its accessibility and simplicity. Although it has been widely used, this metric lacks the complexity and numerous dimensions of research production and effect because it is too basic [57].

Additionally, bibliometric analysis has been used in the research of supply chain management [58–62]. Other researchers have employed bibliometric analysis to study a specific SCM or one single journal or institution, while we focused on supply chain management as a whole through bibliometric analysis by collecting data on the entire range of journals and institutions. When independent researchers or collectives (including supply chain upstream and downstream companies, academia, and government departments) seek partnership partners in a specific area of supply chain, and seek to obtain a concise overview of comprehensive current research hotspots, the lack of relevant intelligence analysis to aid

decision-making often makes the process convoluted and time-consuming. A bibliometric approach can solve the above problems relatively fairly but, at present, scholars lack a comprehensive overview of SCM with this approach, and there has not been a panoramic study of SCM; therefore, the research in this paper is necessary. This paper evaluates the present state and development patterns of supply chain management (SCM) by exposing the contributions of leading nations and regions, the most productive institutions, journals, authors, author keywords, and the most cited publications, through bibliometric analysis. Moreover, we use the bibliometric method to reflect the current research status, hot frontiers, and development trends in the field of supply chain management by analyzing keywords. The following is the structure of this paper. In Section 2, we go through the data sources, search methodologies, and analytical methods. Section 3 contains the descriptions of the results. The discussions are in Section 4, the conclusions are in Section 5, and the future prospects and limitations are presented in Section 6.

## 2. Data Collection and Analysis Methods

A bibliometric analysis approach was adopted and the analysis process can be summarized in the following four parts. The first step was the determination of the search query. We identified search expressions that comprehensively and precisely searched the SCM domain. Thus, the search query was TS = "supply chain management", and Title, Abstract, Author Keywords, and Keywords Plus were included in the search's parameters. Then, we collected the data.

The Web of Science (WoS) core collection was used to retrieve the related documents in supply chain management (SCM). The literature search was conducted on 28 June 2021, using the databases of Science Citation Index Expanded (SCI-EXPANDED) and Social Sciences Citation Index (SSCI). A total of 12,868 papers published from 2010 to 2020 were collected; when the type of literature was reviews and articles, the number was 8998, including 335 highly cited papers and 14 hot papers. Endnote is the industry standard software tool for publishing and managing bibliographies, citations, and references, and all the information for each paper can be stored in Endnote.

The records of 8998 reviews and articles were extracted from WoS to Derwent Data Analyzer (DDA) to manage the data analysis. DDA is a platform for data mining and visualization on desktop computers, and can be used to count the frequency of keywords. We applied DDA to analyze the characteristics of SCM research from different aspects. Cross relationship maps and DDA cluster maps were applied to explain the collaborative relationships between research areas, countries/regions, and institutions, and bubble charts were adopted to show the development trends of research areas, journals, author keywords, and authors in SCM research more intuitively. The next step was data visualization. The final analysis results are presented in visual form, including Tables, Cross-relationships Maps, Bubble Charts, and DDA Cluster Maps.

Overall, we conducted a comprehensive analysis of the SCM field, aiming to identify the most influential studies, determine the topical areas of research, as well as provide insights into current research interests and future prospects. Instead of the subjective presentation of many literature reviews through pure words only, we used data quantification and graphical presentation to help scholars understand more clearly the progress of SCM research and future trends.

## 3. Results

The number of publications and the trends are crucial indications of a discipline's development level. As previously stated, the SCI and SSCI databases provided 8898 articles and reviews to the supply chain management (SCM) research area from 2010 to 2020, of which 335 are highly cited papers and 14 are hot papers, as retrieved on 28 June 2021. The total number of publications by year was correlated with supply chain management (SCM) trends from 2010 to 2020 (Figure 1). Except for 2014, there has been no decrease in overall SCM publications throughout this time period. From 2010 to 2015, the number of

publications remained generally consistent, with a small increase from 543 to 644. In 2016, 739 articles and reviews were published, and the number of publications surpassed 1000 in 2018, with 1443 in 2020. The number of publications produced by China, the USA, and the UK, account for more than half of all worldwide publications. China ranked top with 2385 articles published between 2010 and 2020, followed by the USA with 2234 publications and the UK with 1183. China's publishing trend increased from 121 in 2010 to 433 in 2020, which is similar to the overall trend, and has more than quadrupled over this time period. The number of articles published by USA every year ranked first from 2010 to 2015, but China has since surpassed them, with reductions in 2011, 2013, and 2016, putting USA in the second position with 2234 total publications. From 2010 to 2015, UK publications were below one hundred, then increased to above one hundred in 2016, with 107, and over two hundred in 2020, with 220, placing the UK third in both annual and total publications.

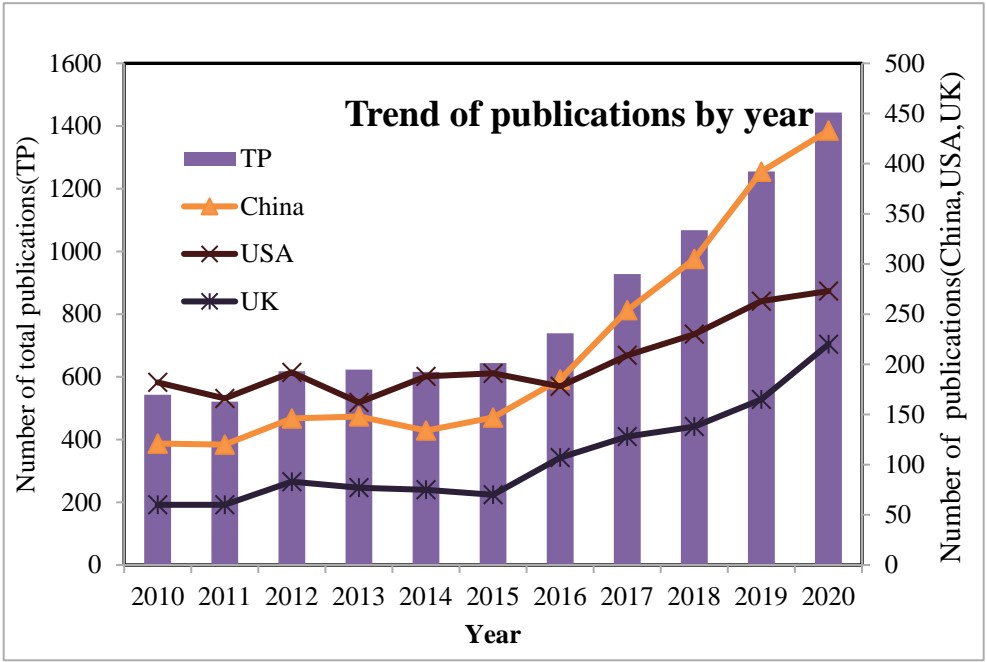

**Figure 1.** Number of publications (China, USA, UK) related to SCM.

### 3.1. Contribution of Leading Research Areas

The research area is one of the information included in each publication, which is classified by Web of Science (WoS) and also known as WoS categories. The 8898 publications on the SCM encompass 153 Web of Science categories since the study fields represent application ranges of the subject. The top 20 WoS research areas in SCM ranked by related total papers (Table 1). "Management" (3071, 34.13), "Operations Research & Management Science" (2680, 29.78), and "Engineering, Industrial" (1854, 20.60) occupied the top three concerned with TP and TPR%. "Engineering, Manufacturing" accounted for 17.47 percent of total papers in the field (TPR%), "Environmental Sciences" with 13.31%, "Business" with 12.04%, and "Green & Sustainable Science & Technology" with 11.46%. The remaining research areas made up less than ten percent of the total. "Engineering, Environmental" dominated the average citations per publication (ACPP) with 50.59. With comparatively high TC (104,075 and 101,978), "Management" and "Operations Research & Management Science" become prominent literature.

**Table 1.** Contribution of the top 20 WoS research areas in SCM.

| Rank | WOS Research Area | TP | TPR% | TC | ACPP |
|---|---|---|---|---|---|
| 1 | Management | 3071 | 34.13 | 104,075 | 33.89 |
| 2 | Operations Research & Management Science | 2680 | 29.78 | 101,978 | 38.05 |
| 3 | Engineering, Industrial | 1854 | 20.60 | 61,353 | 33.09 |
| 4 | Engineering, Manufacturing | 1572 | 17.47 | 54,396 | 34.60 |
| 5 | Environmental Sciences | 1198 | 13.31 | 40,727 | 34.00 |
| 6 | Business | 1083 | 12.04 | 33,529 | 30.96 |
| 7 | Green & Sustainable Science & Technology | 1031 | 11.46 | 35,418 | 34.35 |
| 8 | Engineering, Environmental | 678 | 7.54 | 34,302 | 50.59 |
| 9 | Computer Science, Interdisciplinary Applications | 641 | 7.12 | 15,623 | 24.37 |
| 10 | Environmental Studies | 598 | 6.65 | 10,438 | 17.45 |
| 11 | Computer Science, Artificial Intelligence | 373 | 4.15 | 12,853 | 34.46 |
| 12 | Computer Science, Information Systems | 294 | 3.27 | 6919 | 23.53 |
| 13 | Economics | 262 | 2.91 | 6629 | 25.30 |
| 14 | Engineering, Electrical & Electronic | 256 | 2.85 | 8712 | 34.03 |
| 15 | Engineering, Multidisciplinary | 235 | 2.61 | 4308 | 18.33 |
| 16 | Transportation | 205 | 2.28 | 5845 | 28.51 |
| 17 | Automation & Control Systems | 180 | 2.00 | 3958 | 21.99 |
| 18 | Mathematics, Interdisciplinary Applications | 174 | 1.93 | 3104 | 17.84 |
| 19 | Engineering, Civil | 150 | 1.67 | 4982 | 33.21 |
| 20 | Transportation Science & Technology | 138 | 1.53 | 4960 | 35.94 |

Abbreviations: TP, total papers; TRP%, percent of total papers in the field; TC, total citations; ACPP, average citations per publication.

A bubble chart with years at the top and WoS categories on the left illustrates the development of several study topics through time. Each bubble's number represents the number of particular publications in each WoS study topic, which is proportional to the bubble's size (Figure 2). The top categories for each year may be identified by comparing the size of the bubbles vertically, while the growth trend of each category over time can be determined by comparing the size of the bubbles horizontally. The number of publications remained relatively stable from 2010 to 2016, then gradually climbed from 283 to 425 between 2017 and 2020, for the area of "management," which mostly occupied first place except for the year 2014, when "Operations Research & Management Science" with 226 articles surpassed it. In detail, in the area of "Operations Research & Management Science", 194 and 182 articles were published, respectively, in 2010 and 2011. Since 2012, the number of publications in this field has surpassed 200 every year, peaking at 343 in 2019. Moreover, the development patterns of "Engineering, Industrial" and "Engineering, Manufacturing" were comparable, starting at 119 and 104 in 2010, decreasing relatively in 2011 and increasing significantly in 2012, dropping in the next two years, and then returning to 160 and 154 in 2015, continuing to grow in 2017 and peaking in 2019 with 288 and 213, then finally falling slightly in 2020. Another noteworthy finding refers to the "Environmental Sciences", "Green & Sustainable Science & Technology", "Engineering, Environmental" and "Environmental Studies", all of which had general numbers of publications before 2016, but dramatically surged between 2017 to 2020. The research area of "Environmental Sciences" came in second after "management" with 296 publications in 2020. Therefore, the application of SCM in the environmental research areas has developed qualitatively since 2017. In addition, figures regarding "Business" were relatively high compared with other categories that list after the first four lines from 2010 to 2016, exceeding 100 in 2018 and raising to 190 in 2020.

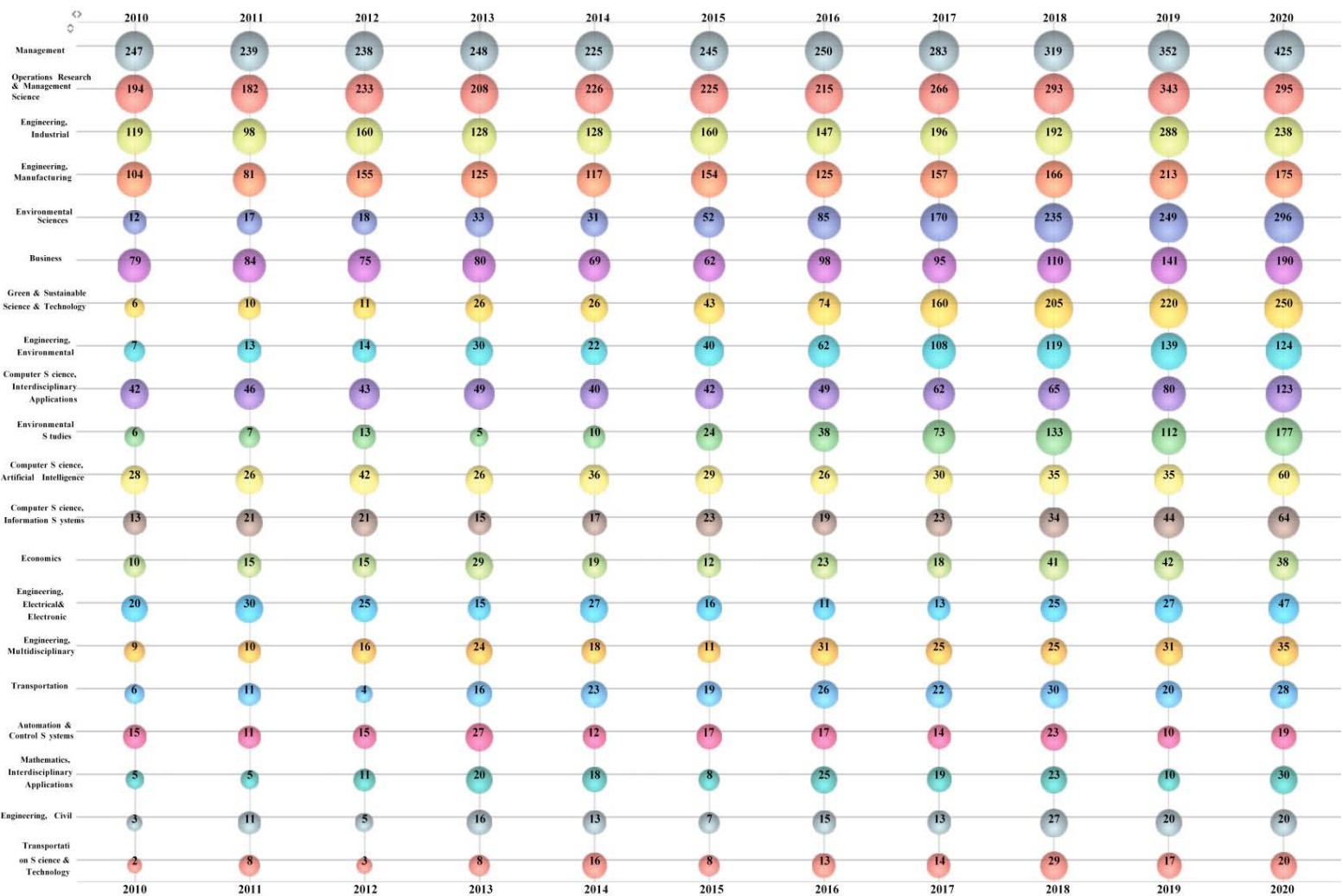

**Figure 2.** Bubble chart of top 20 WoS research areas in SCM.

*3.2. Contribution of Leading Journals*

Clarifying the productive journals publishing articles in SCM is beneficial for researchers to access information and submit manuscripts. The number of TP, TC, ACPP, and IF were listed to conclude the Top 20 journals publishing articles on SCM (Table 2). The journal *J. Clean Prod.* (554, 6.16%), *Int. J. Prod. Econ.* (494, 5.49%), *Int. J. Prod. Res.* (465, 5.17%) and *Eur. J. Oper. Res.* (446, 4.96%) ranked in the top four in terms of the TP published from 2010 to 2020. The most cited journals were *J. Clean Prod.* and *Int. J. Prod. Econ.*, with 28,392 and 27,812 total citations separately. The highest average citations per publication (ACPP) belonged to *Expert Syst. Appl.* with 60.46, and the highest value of impact factor (IF) was 10.302 contributed by *Bus. Strateg. Environ.* Additionally, the TP of *Sustainability* was 350 and the TC of *Supply Chain Manag.* was 13,326. The influence of other journals was relatively low and similar.

A bubble chart was employed to reveal the top 20 productive journals from 2010 to 2020 (Figure 3). The journal *Eur. J. Oper. Res.* published 44 articles in 2010, ranking the first, and fluctuated slightly between 28–55 during this period. This was followed by *Int. J. Prod. Econ.* and *Int. J. Prod. Econ.* with 38 and 36 publications in 2010. The fluctuations were also gentle, except for the year 2012 with a larger growth to 69 articles (*Int. J. Prod. Econ.*) and 2019 with 85 articles (*Int. J. Prod. Econ.*). Concerning the journal *J. Clean Prod.*, five publications occurred in 2010 and continuously increased to 54 in 2016 and exceeded a hundred since 2019. On the other hand, there was only one article published in 2011 of *Sustainability*, three articles in 2014, and a rapid increase since 2017 from 45 to 108, leading to first place in 2020.

**Table 2.** The Top 20 Journals Publishing Articles in Supply Chain Management.

| Rank | Journal Title | TP | TC | ACPP | IF |
|---|---|---|---|---|---|
| 1 | J. Clean Prod. | 554 | 28,392 | 51.25 | 9.297 |
| 2 | Int. J. Prod. Econ. | 494 | 27,812 | 56.30 | 7.885 |
| 3 | Int. J. Prod. Res. | 465 | 13,646 | 29.35 | 8.568 |
| 4 | Eur. J. Oper. Res. | 446 | 18,881 | 42.33 | 5.334 |
| 5 | Sustainability | 350 | 3329 | 9.51 | 3.251 |
| 6 | Supply Chain Manag. | 312 | 13,326 | 42.71 | 9.012 |
| 7 | Int. J. Phys. Distrib. Logist. Manag. | 228 | 9318 | 40.87 | 6.309 |
| 8 | Int. J. Oper. Prod. Manage. | 212 | 7931 | 37.41 | 6.629 |
| 9 | Comput. Ind. Eng. | 193 | 4829 | 25.02 | 5.431 |
| 10 | Int. J. Logist. Manag. | 190 | 3945 | 20.76 | 5.661 |
| 11 | Prod. Plan. Control | 185 | 4019 | 21.72 | 7.044 |
| 12 | Ind. Manage. Data Syst. | 128 | 2956 | 23.09 | 4.224 |
| 13 | J. Supply Chain Manag. | 118 | 6489 | 54.99 | 8.647 |
| 14 | Ann. Oper. Res. | 108 | 2339 | 21.66 | 4.854 |
| 15 | Expert Syst. Appl. | 104 | 6288 | 60.46 | 6.954 |
| 16 | Int. J. Logist.-Res. Appl. | 99 | 1375 | 13.89 | 3.821 |
| 17 | J. Bus. Logist. | 96 | 4471 | 46.57 | 6.677 |
| 18 | Bus. Strateg. Environ. | 89 | 2210 | 24.83 | 10.302 |
| 19 | J. Manuf. Technol. Manag. | 82 | 1660 | 20.24 | 7.547 |
| 20 | Prod. Oper. Manag. | 82 | 2302 | 28.07 | 4.965 |

Abbreviations: TP, total papers; TC, total citations; ACPP, average citations per publication; IF, Impact Factor 2020.

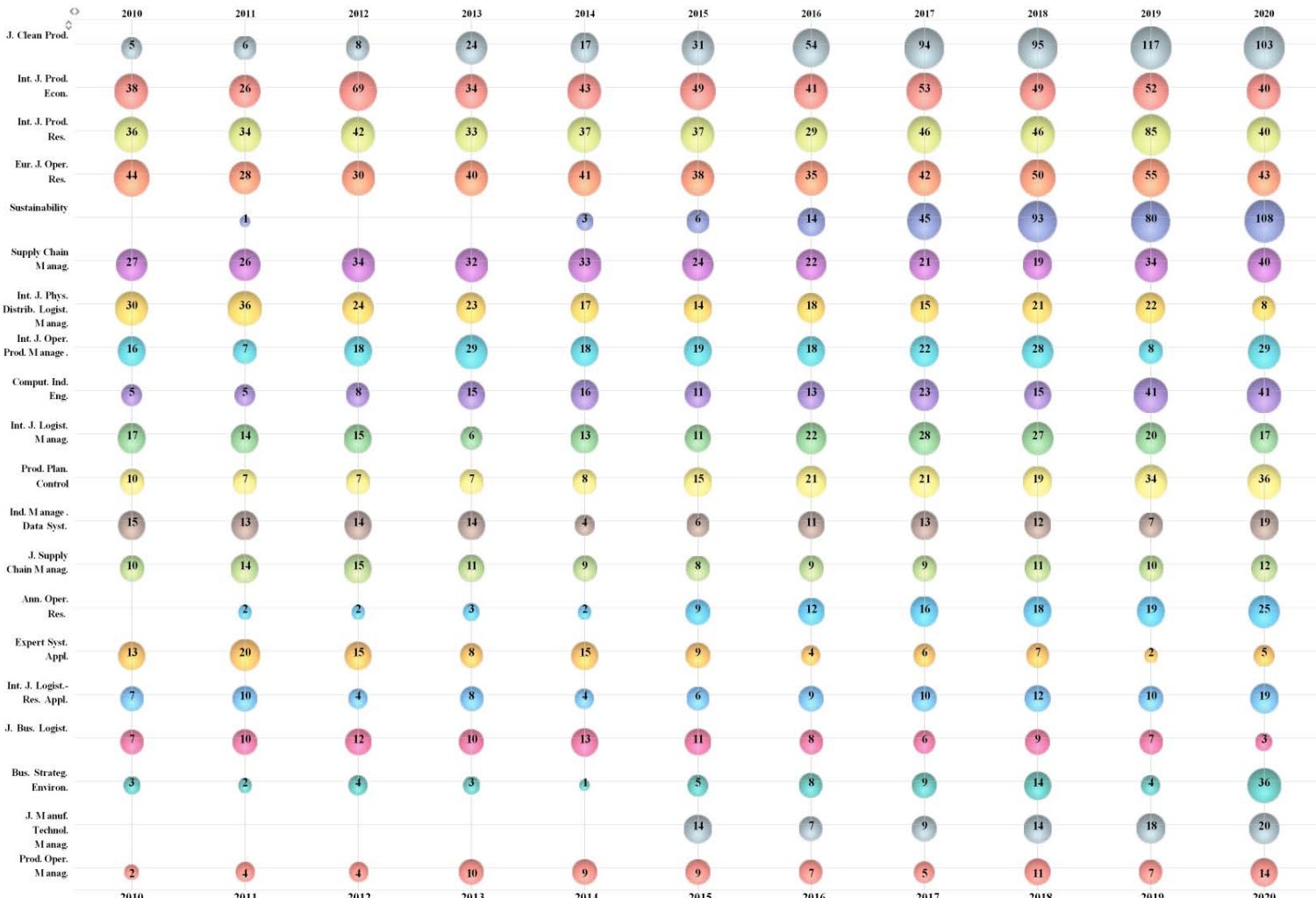

**Figure 3.** Bubble chart of the top 20 productive journals in SCM.

### 3.3. Contribution of Leading Countries/Regions

The most productive country was China with the highest total publications (2385), revealing the highest research influence and attention in SCM (Table 3). China also contributed the highest proportion of global publications of SCM with 26.51% (Figure 4). The USA (2234, 24.83%) and UK (consisting of England, Scotland, Northern Ireland, and Wales) (1183, 13.15%) ranked second and third due to the total publications and proportions. The publications of the top three accounted for more than half of the total proportion, whereas, the highest TC and ACPP belonged to USA (83,663, 37.45), Germany ranked the second with 37.40 of ACPP and UK ranked the third both in TP and TC. Denmark held the highest share of publications (SP) with 83.07, followed by France with 78.98. The USA had the largest number of cooperative countries (nCC) with 88, followed by the UK with 73 cooperative countries. Furthermore, the h-index of the USA reached 125 implying 125 articles had been published with at least 125 citations for each paper. Accordingly, the top three productive countries were China, USA, and UK, from Asia, America, and Europe, respectively (Table 3). Australia was the only nation from Oceania featured in Table 3 and ranked 7th with 398 total publications and a 4.42% proportion. The top 20 countries were from Asia, America, Europe, and Oceania, and half of them were European countries. On the other hand, the collaborative relationships among the top 20 most productive countries/regions identified China as the most active country that had the most collaborations with the USA, UK, and Australia (Figure 5).

**Table 3.** The Top 20 Most Productive Countries/Regions During 2010–2020.

| Rank | Country | TP | TC | ACPP | SP (%) | nCC | H-Index | Region |
|------|---------|-----|------|------|--------|-----|---------|--------|
| 1 | China | 2385 | 64,896 | 27.21 | 42.56 | 59 | 106 | Asia |
| 2 | USA | 2234 | 83,663 | 37.45 | 52.86 | 88 | 125 | Americas |
| 3 | UK | 1183 | 41,781 | 35.32 | 67.46 | 73 | 94 | Europe |
| 4 | India | 585 | 19,432 | 33.22 | 49.74 | 47 | 71 | Asia |
| 5 | Germany | 539 | 20,161 | 37.40 | 52.32 | 46 | 70 | Europe |
| 6 | Iran | 418 | 14,965 | 35.80 | 37.08 | 37 | 57 | Asia |
| 7 | Australia | 398 | 11,464 | 28.80 | 73.37 | 52 | 52 | Oceania |
| 8 | Italy | 390 | 11,716 | 30.04 | 50.51 | 47 | 56 | Europe |
| 9 | France | 385 | 11,275 | 29.29 | 78.96 | 57 | 57 | Europe |
| 10 | Spain | 373 | 10,867 | 29.13 | 56.57 | 47 | 52 | Europe |
| 11 | Canada | 370 | 12,889 | 34.84 | 72.16 | 51 | 56 | Americas |
| 12 | South Korea | 310 | 5996 | 19.34 | 47.42 | 25 | 40 | Asia |
| 13 | Netherlands | 279 | 8250 | 29.57 | 61.29 | 43 | 47 | Europe |
| 14 | Brazil | 264 | 7063 | 26.75 | 49.62 | 36 | 45 | Americas |
| 15 | Sweden | 210 | 6289 | 29.95 | 53.33 | 35 | 44 | Europe |
| 16 | Turkey | 203 | 4927 | 24.27 | 33.50 | 34 | 39 | Europe |
| 17 | Denmark | 189 | 14,080 | 74.50 | 83.07 | 33 | 64 | Europe |
| 18 | Malaysia | 186 | 7344 | 39.48 | 72.58 | 39 | 42 | Asia |
| 19 | Finland | 176 | 4680 | 26.59 | 55.68 | 38 | 37 | Europe |
| 20 | Switzerland | 129 | 4670 | 36.20 | 72.09 | 33 | 35 | Europe |

Abbreviations: TP, total papers; TC, total citations; ACPP, average citations per publication; SP, share of publications; nCC, number of cooperative countries. Note: The statistics for Taiwan are included in China's.

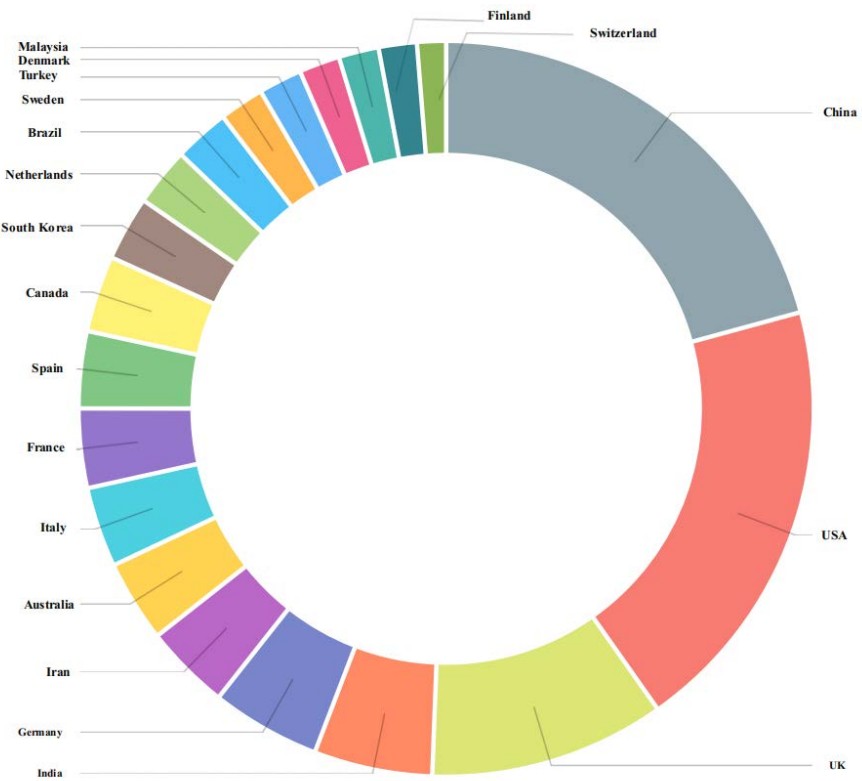

**Figure 4.** Pie chart of the top 20 countries/regions for publications in SCM.

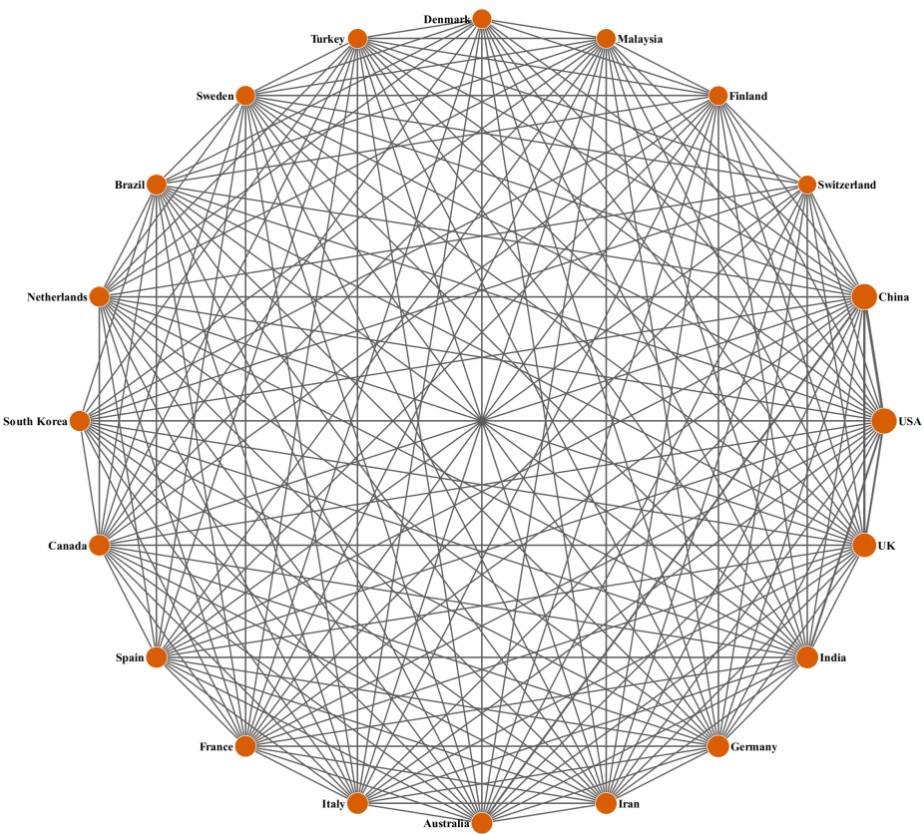

**Figure 5.** Collaboration matrix map among the top 20 most productive countries/regions.

### 3.4. Contribution of Leading Institutions

The statistics of TP, TC, and h-index for the top 20 most productive institutions can provide researchers with specific information in detail (Table 4). The wide influence of Hong Kong Polytech Univ was shown in both the highest number of publications and citations in total (238, 12,490), which were far ahead of other institutions in terms of TP and TC. Islamic Azad University and University of Tennessee were 2nd and 3rd with 135 and 107 total papers, respectively. The University of Southern Denmark had the highest ACCP with 105.31 and ranked second in TC with 8741. The value of ACCP of the University of Kassel from Germany was 82.94, ranking after Univ Southern Denmark. For the h-index, Hong Kong Polytech University, the University of Southern Denmark, and the University of Tennessee occupied the top three (61, 49, 42). Consequently, these institutions, whose locations are all from the top 20 countries/region, play an important role in developing and promoting SCM research. As mentioned with the development trend of publications of the top 20 institutions, Hong Kong Polytech University was the most productive institutions in almost each year (except 2017, see Figure 6). Hong Kong Polytech University suffered with three declines and peaked in 2018 with 31. The publications of Islamic Azad Univ increased significantly in 2014 with 11, and grew steadily since then. The University of Southern Denmark started publications of SCM in 2013 and increased the number in the following years, but a sudden and sharp decline appeared after 2019. It is also worth noting that Montpellier Business School published none until 2017 and expanded rapidly in the next few years. In terms of collaborative relationships among the top 20 most productive institutions, Hong Kong Polytech University was also the most active institutions of collaboration and had a closest relationship with Dalian University of Technology (Figure 7).

**Table 4.** The top 20 most productive institutions of publications during 2010–2020.

| Rank | Institution | TP | TC | ACCP | H-Index | Country |
|------|-------------|----|----|------|---------|---------|
| 1 | Hong Kong Polytech Univ | 238 | 12,490 | 52.48 | 61 | China |
| 2 | Islamic Azad Univ | 135 | 4411 | 32.67 | 35 | Iran |
| 3 | Univ Tennessee | 107 | 5372 | 50.21 | 42 | USA |
| 4 | Michigan State Univ | 98 | 3754 | 38.31 | 33 | USA |
| 5 | Arizona State Univ | 86 | 4265 | 49.59 | 32 | USA |
| 6 | Univ Southern Denmark | 83 | 8741 | 105.31 | 49 | Denmark |
| 7 | Univ Nottingham | 81 | 2589 | 31.96 | 29 | UK |
| 8 | Univ Tehran | 81 | 2624 | 32.40 | 29 | Iran |
| 9 | Dalian Univ Technol | 80 | 4762 | 59.53 | 33 | China |
| 10 | Politecn Milan | 79 | 2599 | 32.90 | 28 | Italy |
| 11 | Cardiff Univ | 76 | 3033 | 39.91 | 30 | UK |
| 12 | Tianjin Univ | 72 | 1296 | 18.00 | 19 | China |
| 13 | Montpellier Business Sch | 68 | 1877 | 27.60 | 28 | France |
| 14 | Shanghai Jiao Tong Univ | 68 | 1617 | 23.78 | 24 | China |
| 15 | Indian Inst Technol | 67 | 2367 | 35.33 | 27 | India |
| 16 | Natl Taiwan Univ Sci & Technol | 63 | 1743 | 27.67 | 20 | Taiwan region |
| 17 | Univ Kassel | 62 | 5142 | 82.94 | 31 | Germany |
| 18 | Auburn Univ | 60 | 2202 | 36.70 | 26 | USA |
| 19 | Univ Arkansas | 60 | 2148 | 35.80 | 22 | USA |
| 20 | Univ Elect Sci & Technol China | 60 | 1600 | 26.67 | 24 | China |

Abbreviations: TP, total papers; TC, total citations; ACPP, average citations per publication.

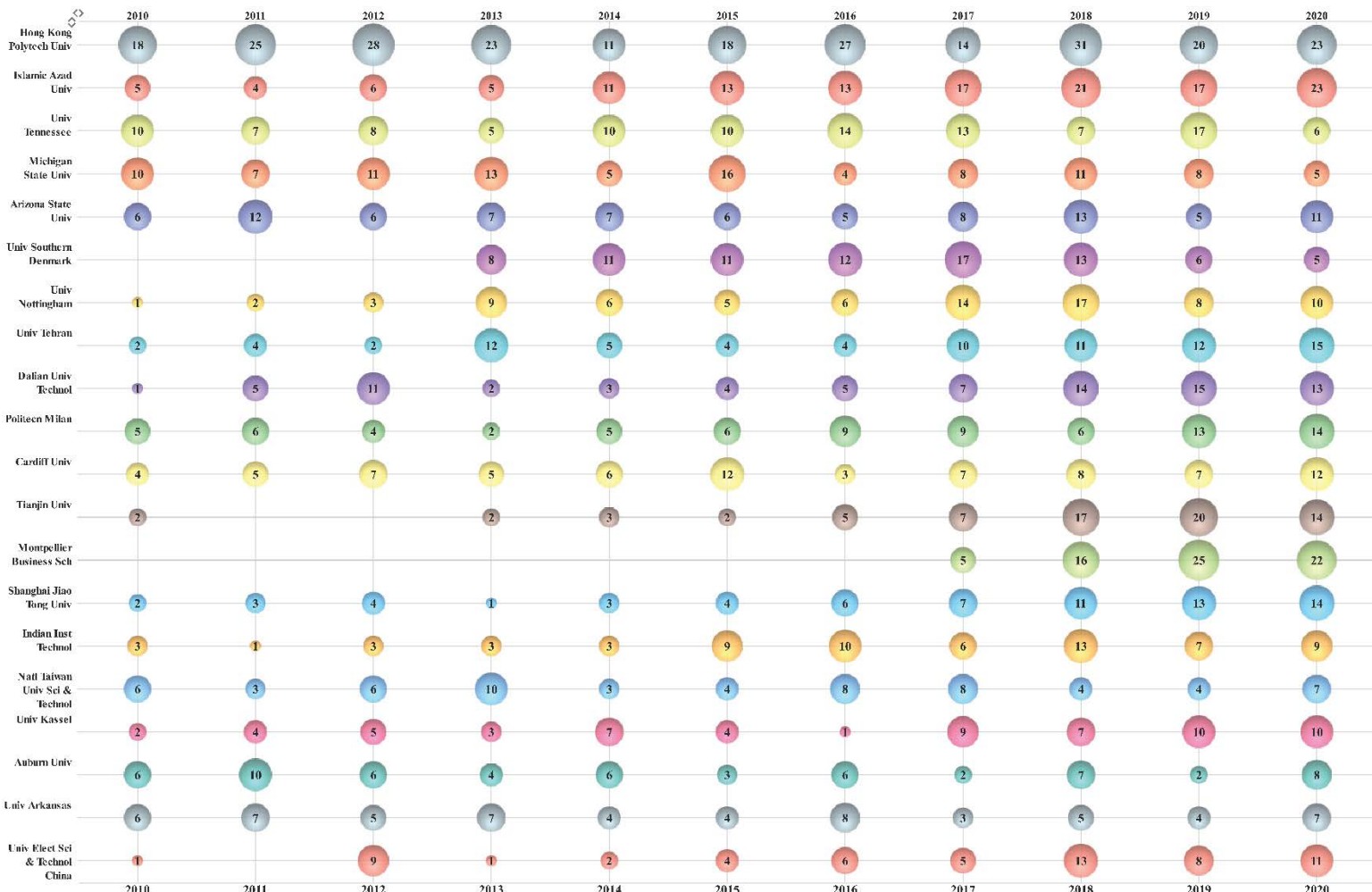

**Figure 6.** Bubble chart of top 20 productive journals institutions.

### 3.5. Leading Authors and Corresponding Authors Who Contributed to the SCM

The top three most productive authors in SCM research according to the total publications were Sarkis J, Govindan K, and Gunasekaran A during these periods (Table 5). Sarkis J contributed the most publications with 78 TP while Govindan K was responsible for the most articles with 47, and possessed the highest value of TC, ACPP, and h-index (9469, 124.59, 50). Choi TM ranked the second of TAR with 43 articles. The top 20 authors are mostly from the top 20 most productive countries/regions, with authors from USA, Denmark, Hong Kong, and China contributing the most. From the perspective of the corresponding author, the top three corresponding authors were Govindan K with 47 articles published, Choi TM with 43 TP, and Chiappetta JCJ with 34 TP (Table 6). Thirteen of the top 20 writers and corresponding authors were identical when compared (Tables 5 and 6). They were Sarkis J, Govindan K, Gunasekaran A, Choi TM, Tseng ML, Seuring S, Mangla SK, Luthra S, Sarkar B, Xiao TJ, Zhu QH, Saen RF, and Chen X, who made a substantial contribution to the supply chain management research. Specifically, Chiappetta JCJ was not among the top 20 authors, while Govindan K was second and Choi TM was fourth of the top 20 authors. With the exception of Li Y, Hazen BT, Kumar S, Schoenherr T, De Giovanni P, and Huo BF, who are not included in Table 5, the remaining corresponding authors are mostly the same as those in the top 20 authors list. The corresponding authors ranked fourth to tenth in Table 6 matched the list of the top 20 authors in Table 5.

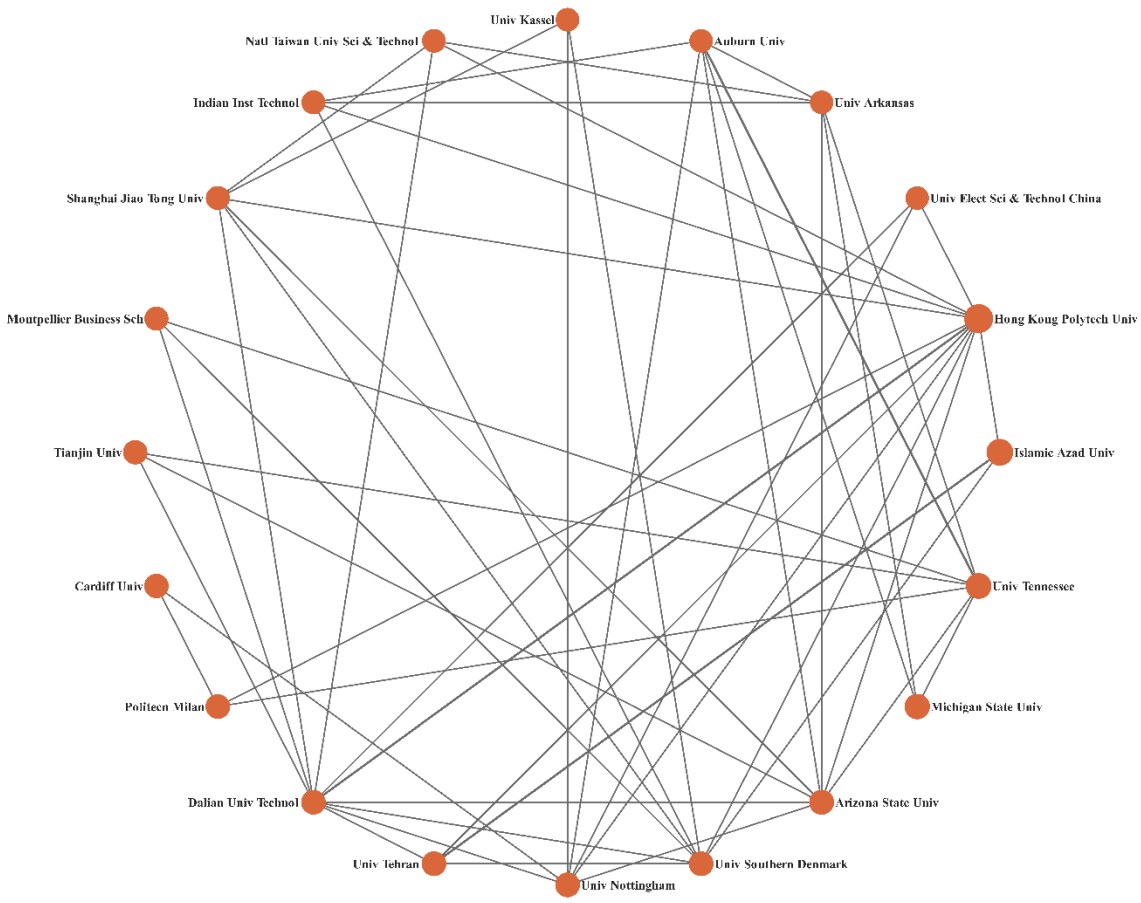

**Figure 7.** Collaboration matrix map among the top 20 most productive institutions of publications.

**Table 5.** Contribution of the top 20 authors in SCM.

| Rank | Author | TP | TAR | TC | ACPP | H-Index | Institution(Current), Country/Region |
|---|---|---|---|---|---|---|---|
| 1 | Sarkis J | 78 | 18 | 7926 | 101.62 | 41 | Worcester Polytech Inst, USA |
| 2 | Govindan K | 76 | 47 | 9469 | 124.59 | 50 | Univ Southern Denmark, Denmark |
| 3 | Gunasekaran A | 69 | 33 | 5088 | 73.74 | 40 | Calif State Univ, USA |
| 4 | Choi TM | 55 | 43 | 2571 | 46.75 | 29 | Hong Kong Polytech Univ, Hong Kong, China |
| 5 | Jabbour CJC | 50 | 34 | 2325 | 46.50 | 26 | Montpellier Business Sch, France |
| 6 | Tseng ML | 42 | 28 | 1879 | 44.74 | 21 | Asia Univ, Taiwan, China |
| 7 | Cheng TCE | 40 | 3 | 1705 | 42.63 | 25 | Hong Kong Polytech Univ, Hong Kong, China |
| 8 | Jabbour ABLD | 40 | 5 | 1930 | 48.25 | 22 | Univ Lincoln, England |
| 9 | Seuring S | 40 | 16 | 4442 | 111.05 | 26 | Univ Kassel, Germany |
| 10 | Mangla SK | 39 | 16 | 1637 | 41.97 | 24 | Univ Plymouth, England |
| 11 | Luthra S | 37 | 17 | 1840 | 49.73 | 24 | Govt Polytech, India |
| 12 | Sarkar B | 36 | 26 | 938 | 26.06 | 18 | Yonsei Univ, South Korea |
| 13 | Xiao TJ | 34 | 23 | 854 | 25.12 | 18 | Nanjing Univ, China |
| 14 | Zhu QH | 34 | 23 | 3410 | 100.29 | 24 | Shanghai Jiao Tong Univ, China |
| 15 | Chan FTS | 31 | 13 | 1149 | 37.06 | 18 | Hong Kong Polytech Univ, Hong Kong, China |
| 16 | Saen RF | 31 | 26 | 902 | 29.10 | 14 | Sohar Univ, Oman |
| 17 | Dubey R | 30 | 11 | 1967 | 65.57 | 25 | Montpellier Business Sch, France |
| 18 | Lai KH | 29 | 5 | 3083 | 106.31 | 23 | Hong Kong Polytech Univ, Hong Kong, China |
| 19 | Papadopoulos T | 29 | 6 | 2223 | 76.66 | 26 | Univ Kent, England |
| 20 | Chen X | 28 | 19 | 1046 | 37.36 | 17 | Univ Elect Sci & Technol China, China |

Abbreviations: TP, total papers; TAR, total articles he/she is responsible for; TC, total citations; ACPP, average citations per publication.

**Table 6.** Contribution of the top 20 corresponding authors in SCM.

| Rank | Author | TP | TC | ACPP | H-Index | Institution(Current), Country/Region |
|---|---|---|---|---|---|---|
| 1 | Govindan, Kannan | 47 | 7516 | 159.91 | 42 | Univ Southern Denmark, Denmark |
| 2 | Choi, Tsan-Ming | 43 | 2307 | 53.65 | 29 | Hong Kong Polytech Univ, Hong Kong, China |
| 3 | Chiappetta Jabbour, Charbel Jose | 34 | 2121 | 62.38 | 24 | EMLYON Business Sch, France |
| 4 | Gunasekaran, Angappa | 33 | 3324 | 100.73 | 28 | Calif State Univ, USA |
| 5 | Tseng, Ming-Lang | 28 | 1547 | 55.25 | 17 | Asia Univ, Taiwan, China |
| 6 | Saen, Reza Farzipoor | 26 | 908 | 34.92 | 15 | Sohar Univ, Oman |
| 7 | Sarkar, Biswajit | 26 | 766 | 29.46 | 15 | Yonsei Univ, South Korea |
| 8 | Zhu, Qinghua | 23 | 2046 | 88.96 | 19 | Shanghai Jiao Tong Univ, China |
| 9 | Chen, Xu | 22 | 859 | 39.05 | 16 | Univ Elect Sci & Technol China, China |
| 10 | Xiao, Tiaojun | 22 | 649 | 29.5 | 13 | Nanjing Univ, China |
| 11 | Li, Yongjian | 19 | 889 | 46.79 | 15 | Nankai Univ, China |
| 12 | Sarkis, Joseph | 18 | 2224 | 123.56 | 15 | Worcester Polytech Inst, USA |
| 13 | Luthra, Sunil | 17 | 1388 | 81.65 | 15 | Ch Ranbir Singh State Inst Engn & Technol, India |
| 14 | Hazen, Benjamin T. | 16 | 1130 | 70.63 | 12 | Air Force Inst Technol, USA |
| 15 | Mangla, Sachin Kumar | 16 | 766 | 47.88 | 12 | Univ Plymouth, UK |
| 16 | Kumar, Sameer | 16 | 734 | 45.88 | 9 | Univ St Thomas, USA |
| 17 | Schoenherr, Tobias | 16 | 950 | 59.38 | 14 | Michigan State Univ, USA |
| 18 | Seuring, Stefan | 16 | 2400 | 150 | 14 | Univ Kassel, Germany |
| 19 | De Giovanni, P | 14 | 564 | 40.29 | 11 | LUISS Univ, Italy |
| 20 | Huo, Baofeng | 14 | 2141 | 152.93 | 9 | Tianjin Univ, China |

*3.6. Analysis of Yearly Most Cited Papers*

Analyzing citation frequency of a paper can reveal its significance in the research field, despite the fact that numerous variables influence the citation impact. The most cited article related to SCM by year was "The impact of supply chain integration on performance: A contingency and configuration approach" with 1235 total citations, which was published by *Int. J. Prod. Econ.* in 2010 (Table 7). Flynn et al. [63] added to the body of knowledge on supply chain integration (SCI), which correlated with operational and

economic performance. The second most cited paper, "An organizational theoretic survey of green supply chain management literature", focused on exploring new directions and identifying future directions of green supply chain management (GSCM) [64]. Govindan et al. [65] published "Reverse logistics and closed-loop supply chain: A comprehensive review to explore the future" to review the reverse logistic and closed-loop supply chain in scientific journals, ranking in third position in TC. The article "A state-of-the-art survey of TOPSIS applications" ranked fourth with 809 citations. Behzadian et al. [66] developed the Technique for Order Preference by Similarity to Ideal Solution (TOPSIS) to classify the research on TOPSIS applications and methodologies. The paper "Literature review of Industry 4.0 and related technologies", which studied the characteristics and content of Industry 4.0 for enterprises [67], had the highest 210 total citations per year (TCY). The most cited paper in 2017 also provided a review of Industry 4.0 [32]. The paper titled "Blockchain technology and its relationships to sustainable supply chain management" ranked second in TCY with 193 citations per year. Inter-organizational, intra-organizational, technical, and external barriers were introduced as four categories of barriers to the use of blockchain technology. Future research proposals and ways to get beyond these barriers were also presented [68]. The most cited paper in 2018 also shared early data showing how using blockchain in supply chain operations might improve accountability and transparency [69]. Besides, Ahi et al. [70] discovered and reviewed existing definitions of sustainable supply chain management (SSCM) and green supply chain management (GSCM). Brandenburg et al. [71] offered a content analysis of 134 carefully chosen works on formal, quantitative models that tackle sustainability issues in the future SC. The most cited article in 2016 emphasized the application of big data in SCM [72].

*3.7. Analysis of Author Keywords*

Author keywords based on the numbers of specific keywords used were introduced for analysis of the trend of research as they provide further information about the study topics. A bubble chart of top author keywords can determine the trends and recent hot issues, and allow the quick visual identification of pattern changes [73]. The author keywords, year of publication, and the number of publications are three aspects of the data that the bubble chart displays (Figure 8). In addition, we applied data cleaning to ensure that keywords with the same meaning are represented by a uniform word.

From 2010 to 2020, 14,723 author keywords were utilized to examine the primary concerns of authors and the research trend. There were terms used only once, accounting for around 73.2 percent of all, indicating that SCM research drew widespread interest. The top 35 author keywords by year depicted "Supply chain (management)" dominated the total number of times used from 2010 to 2020, with 4112 times. Besides, "Sustainable development/ (Environmental) sustainability" was the second most often used keyword, with 823 instances, increasing steadily from 14 to 194 instances. With 423 and 363 searches, the terms "Green supply chain (management)" and "Sustainable supply chain (management)" came in third and fourth, respectively. The following keywords were "Systematic literature review/Literature review" (mentioned 286 times), "Game theory" (mentioned 252 times), "Performance/Performance measurement" (mentioned 230 times), and "Inventory/Inventory management" (mentioned 209 times), which all exceeded 200 times in the total record. Moreover, the recorded numbers of "Collaboration/Coordination", "Logistics", "Case study", "Supplier selection", "Structural equation modeling", "Risk management" and "Reverse logistics" were relatively high (194, 192, 183, 176, 149, 145, 133 times). It is also worth noting that "China" was the only keyword in SCM that appeared as a country, appearing 125 times and ranked sixteenth among the top 35 keywords. Since 2018, five publications containing the keyword "block chain" have been published, with the number of papers published drastically increasing over the next two years. The use of "circular economy" in SCM was first presented in 2010 and 2011, then ignored from 2012 to 2016, before being revived in 2017.

**Table 7.** Yearly most cited publications during the period of 2010–2020 [32,63–72].

| Year | Authors | Title | TC | TCY | Source | Country/Region |
|---|---|---|---|---|---|---|
| 2010 | Flynn, BB. et al. | The impact of supply chain integration on performance: A contingency and configuration approach | 1235 | 112 | J. Oper. Manag. | China |
| 2011 | Sarkis, J. et al. | An organizational theoretic review of green supply chain management literature | 918 | 92 | Int. J. Prod. Econ. | Hong Kong, China |
| 2012 | Behzadian, M. et al. | A state-of the-art survey of TOPSIS applications | 809 | 90 | Expert Syst. Appl. | Iran |
| 2013 | Ahi, P. et al. | A comparative literature analysis of definitions for green and sustainable supply chain management | 547 | 68 | J. Clean Prod. | Canada |
| 2014 | Brandenburg, M. et al. | Quantitative models for sustainable supply chain management: Developments and directions | 580 | 83 | Eur. J. Oper. Res. | Germany |
| 2015 | Govindan, K. et al. | Reverse logistics and closed-loop supply chain: A comprehensive review to explore the future | 832 | 139 | Eur. J. Oper. Res. | Denmark |
| 2016 | Wang, G. et al. | Big data analytics in logistics and supply chain management: Certain investigations for research and applications | 441 | 88 | Int. J. Prod. Econ. | USA |
| 2017 | Zhong, RY. et al. | Intelligent Manufacturing in the Context of Industry 4.0: A Review | 591 | 148 | Engineering | New Zealand |
| 2018 | Kshetri, N | Blockchain's roles in meeting key supply chain management objectives | 352 | 117 | Int. J. Inf. Manage. | USA |
| 2019 | Saberi, S. et al. | Blockchain technology and its relationships to sustainable supply chain management | 386 | 193 | Int. J. Prod. Res. | USA |
| 2020 | Oztemel, E. et al. | Literature review of Industry 4.0 and related technologies | 210 | 210 | J. Intell. Manuf. | Turkey |

TC, total citations; TCY, total citations per year.

The use times of the author's keyword for the recent three years (2018–2020) would provide a better understanding of the recent hot trends (Table 8). Aside from ranking first for the keyword supply chain management, sustainability (95, 108, 149 times), sustainable supply chain (59, 89, 85 times), and green supply chain management (59, 58, 77 times) were consistently in the top four. Furthermore, the keywords game theory and sustainable development have appeared regularly in the last three years, with a relatively high ranking. In 2018, the keyword blockchain was only cited five times, ranking 111th, but it swiftly surged to 12th in 2019 with 22 times and fifth with 74 times in 2020. Industry 4.0 was mentioned 13 times in 2018, ranking 23rd, however, jumping to sixth both in 2019 and 2020.

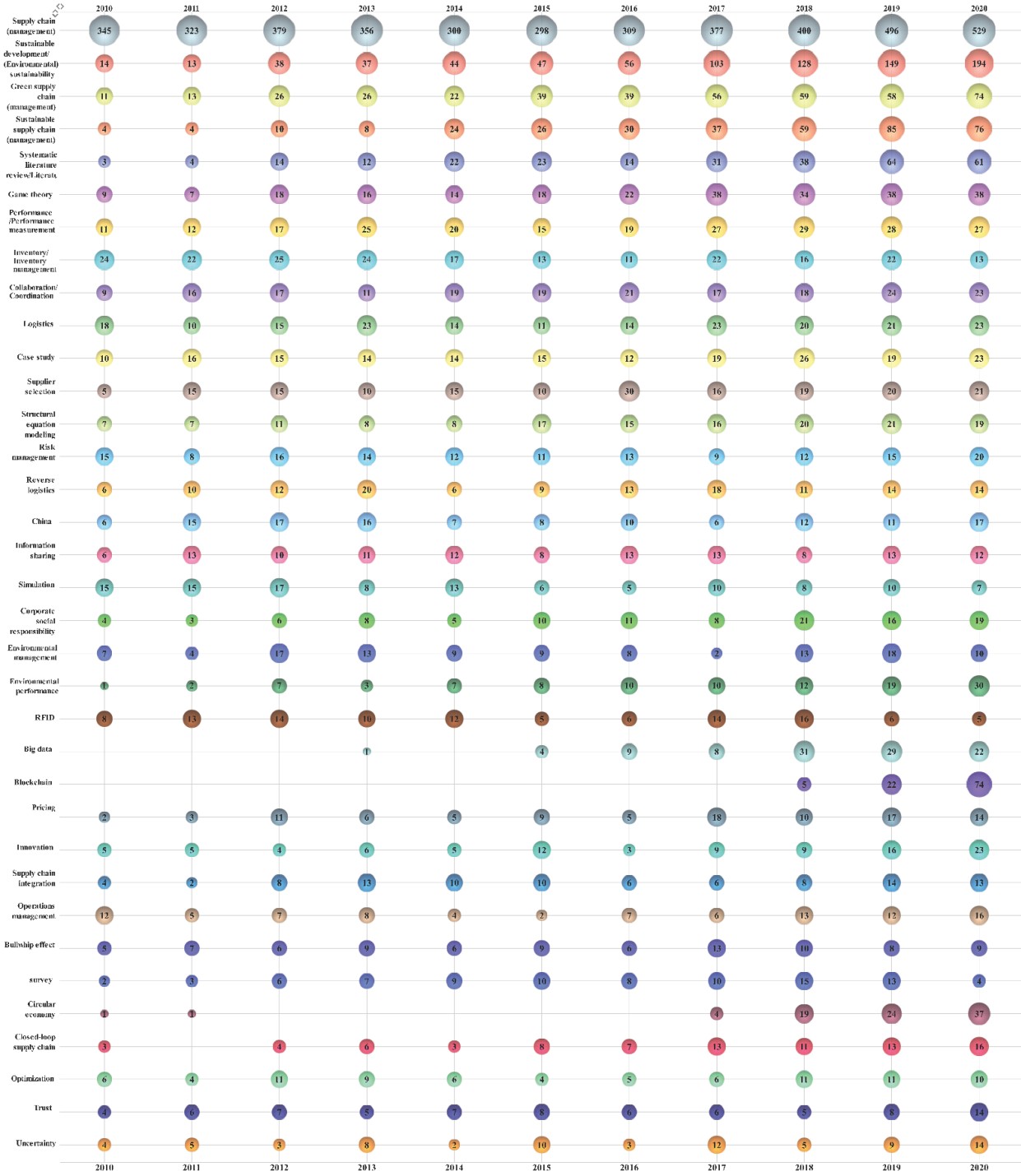

**Figure 8.** Bubble chart of top 35 author keywords by year.

**Table 8.** Top 20 author keywords in the last three years.

| Rank | 2020 | | 2019 | | 2018 | |
|---|---|---|---|---|---|---|
| | Used Times | Author Keywords | Used Times | Author Keywords | Used Times | Author Keywords |
| 1 | 529 | Supply chain management | 496 | Supply chain management | 409 | Supply chain management |
| 2 | 149 | Sustainability | 108 | Sustainability | 95 | Sustainability |
| 3 | 85 | sustainable supply chain management | 89 | sustainable supply chain management | 59 | Green supply chain management |
| 4 | 77 | Green supply chain management | 58 | Green supply chain management | 59 | sustainable supply chain management |
| 5 | 74 | blockchain | 38 | Game theory | 45 | big data |
| 6 | 53 | Industry 4.0 | 33 | Industry 4.0 | 34 | Game theory |
| 7 | 38 | Game theory | 32 | literature review | 29 | Performance measurement |
| 8 | 37 | Circular economy | 32 | Systematic literature review | 26 | Case study |
| 9 | 36 | sustainable development | 31 | sustainable development | 24 | sustainable development |
| 10 | 31 | Systematic literature review | 29 | big data | 23 | corporate social responsibility |
| 11 | 30 | Environmental performance | 24 | Circular economy | 22 | structural equation modeling |
| 12 | 30 | literature review | 22 | blockchain | 20 | literature review |
| 13 | 24 | corporate social responsibility | 21 | Logistics | 20 | Logistics |
| 14 | 23 | Case study | 21 | structural equation modeling | 19 | Circular economy |
| 15 | 23 | innovation | 20 | Supplier selection | 19 | Supplier selection |
| 16 | 23 | Logistics | 19 | Case study | 18 | Systematic literature review |
| 17 | 22 | big data | 19 | Environmental performance | 16 | RFID |
| 18 | 21 | DEMATEL | 18 | Environmental management | 15 | DEMATEL |
| 19 | 21 | Supplier selection | 17 | pricing | 15 | survey |
| 20 | 20 | Risk management | 16 | corporate social responsibility | 14 | Closed-loop supply chain |

## 4. Discussion

A total of 8998 articles and reviews were evaluated to show the expanding content and shifting focus in SCM research from 2010 to 2020. The research results consist of research areas, leading countries and regions, most productive institutions, journals, authors, author keywords, and most cited publications. The data were organized in tables and pictures, such as the number of papers and cooperative countries, total citations, h-index, and percentage of international cooperation. With the exception of supply chain management, Management, Operations Research & Management Science, and Engineering/Industrial were the main research topics, demonstrating how broadly SCM had been implemented

in the management of operations, science, engineering, and industry. As a result, SCM is particularly effective in solving engineering and environmental challenges, as well as the management and operations. The cooperation between different research fields provides more space for supply chain management process research. Additionally, Hong Kong Polytech University was the most active institution of collaboration, and the wide influence attributed both the highest number of publications and citations, which were far ahead of other institutions in terms of TP and TC. China, the USA, and UK contributed the most publications, accounting for more than half of the total proportion. The USA cooperated with China and the UK frequently, and the main collaborative objects of UK were also China and the USA. It was worth noting that none of the top 20 contributing countries or regions were from Africa. Concerning the leading authors and corresponding authors, the most contributing authors were from USA and Denmark while the majority of the top 20 corresponding authors came from China and the USA. The collaboration of scholars from different countries and institutions can jointly promote the progress of research. Moreover, the most cited article by year clearly reflected the hot issue of turning to green and sustainable supply chain management (in the year of 2011, 2013, 2014), reversed and closed-loop SC (in 2015), big data (in 2016), blockchain (in 2018, 2019), and Industry 4.0 (in 2017, 2020).

In detail, the most cited published articles each year during this period focused on sustainability and green supply chain management (GSCM) provided a background discussion on GSCM/SSCM, and reviewed recent literature [64,70,71]. The most cited hot article of 2016 focused on big data and recognized the importance of big data business analytics (BDBA). This reviewed and categorized the literature on BDBA's application in logistics and supply chain management [71]. From 2017 to 2020, researchers preferred to study the application of blockchain and Industry 4.0 in supply chain management research [32,67–69].

The leading author keywords revealed that sustainable development and green supply chain were persistent hot topics from 2010 to 2020, while big data and block chain were emerging hot topic that have attracted the interest of scholars in recent years. When the top author keywords from Figure 8 and Table 8 were merged, we found that sustainability, sustainable supply chain, and green supply chain management were the most popular subjects, with the exception of supply chain management. Sustainability and sustainable supply chain management (SSCM) represent an evolving field of SCM. Carter and Easton [74] performed a systematic evaluation of the literature on sustainable supply chain (SSCM) in the major logistics and supply chain management publications. Brandenburg et al. [71] presented a content analysis of 134 carefully selected works on quantitative, formal models that handle sustainability elements in the forward SC to assess trends and directions in this research field. While several models have been used, life-cycle assessment-based techniques and impact criteria obviously dominated the environment. Seuring [75] considered the social aspect of sustainability was ignored. Ashby et al. [76] reviewed the literature on SCM and found researchers mainly focused on interactions, relationships, and communication, whereas the social dimension of SSCM was recognized but received less attention than expected. The combination of circular economy ideas with sustainable supply chain management might result in considerable environmental advantages [77].

The area of environmental protection is also highly related to SCM research. Environmentally sustainable options are becoming more important in supply-chain management research and practice. Testa and Iraldo [78] used the keyword "environmental performance" to describe the implementation determinants and motivations of green supply chain management (GSCM). It was critical for manufacturers to coordinate internal and external components of GSCM implementation in order to enjoy the performance gains [79]. Green et al. [80] found that industrial firms that use GSCM techniques increase their environmental and economic performance, which has a beneficial influence on operational performance. Kannan D et al. [81] conducted a sensitivity analysis to investigate the impact of decision

makers' preferences for the specified GSCM procedures on the selection of green suppliers. Considering the green supply chain in terms of manufacturing in a certain nation, various automotive component manufacturing businesses in India have distinct challenges when to adopting GSCM. Supplier obstacles, on the other hand, were the most important in their GSCM implementation, particularly in terms of environmental awareness [82].

The application of SCM in the field of information technology and intelligence is rapidly developing as evidence by the key words blockchain and Industry 4.0. In the context of Industry 4.0, which refers to the digitization of industry, the research of intelligent supply chain management driven by new technologies includes block chain technology-driven and big data analysis technology-driven. Treiblmaier [83] submitted a theoretical study that was first published to analyze the relationship and bridge the gap between block chain and SCM. Meng et al. [84] investigated block chain intrusion detection, which can be used in a variety of industries, including SCM. Galvez et al. [85] used block chain technology to validate food supply chain traceability and authenticity. Blossey G et al. [86] provided an overview of the state of the art and identify areas for further study on the use of blockchain technology in SCM. Block chain can transform the practice of operations and supply chain management, including enhancing product safety and security; improving quality management; reducing illegal counterfeiting; improving sustainable supply chain management; advancing inventory management and replenishment, reducing the need for intermediaries; impacting new product design and development, and reducing the cost of supply chain transactions [87]. On the other hand, we are generating massive data every second with the development of the Internet as we all produce and depend on data. Big data became a buzzword in diverse areas, including SCM [88–91]. When Waller and Fawcett [92] studied how supply chain management (SCM) intersects with DPB (data science, predictive analytics, and big data) for the first time in 2013, they predicted the growing popularity of SCM and education. Chen et al. [93] adopted the dynamic capabilities theory to conceive big data analysis usage as a distinct information processing capacity that provides firms with a competitive edge. Kache and Seuring [94] highlighted 43 opportunities and problems related to the advent of Big Data Analytics from a corporate and supply chain viewpoint. In addition, game theory has become an indispensable tool for analyzing supply networks involving several individuals, many of whom have conflicting goals [95]. Tian et al. [96] examined evolutionary game theory to analyze the connections between participants such as the government, businesses, and consumers of green supply chain management (GSCM) in China. There were also considerations for potential game theory applications in SCM [97]. Furthermore, the keyword "performance" referred to both economic performance and environmental performance that could be enhanced by green supply chain management [98,99], and inventory management also had been cited as one of the keywords of supply chain management, as Belien [100] presented a review of the literature on inventory and supply chain management of blood products.

## 5. Conclusions

Theoretical research of supply chain management (SCM) is maturing, while modern supply chain management research that can highlight the demands of new social and economic development and represent the development of human science and technology is booming. It is urgent to integrate new perspectives, theories, and methodologies in the process of cross-fertilization between supply chain management theory and other subjects to constantly enrich theoretical research and the application practice of SCM.

This paper comprehensively collects, analyzes, and reviews various research areas of SCM from 2010–2020. By collecting data from a variety of publications and institutions, we focused on SCM as a whole through the methodology of bibliometric analysis and visualized the data using DDA techniques to help researchers better understand the current situation and the emerging trends of SCM. We also found some interesting details, such as the collaboration between regions, and the network of research relationships between institutions and scholars.

### 6. Future Prospects and Limitations

We summarize the future research directions of SCM that can be undertaken in the following domains based on the trend of highly cited hot papers and the author keywords in recent years.

(1) Environmental perspective of supply chain management (SCM). Governments around the world have established relevant regulations and policies to relieve environmental pressures when facing global eco-environmental challenges. In recent years, many scholars have begun to study SCM from an environmental perspective in the academic field, with a particular emphasis on green supply chain management (GSCM), sustainable supply chain management (SSCM), and low-carbon supply chain management (LSCM) [77,101–104]. According to the ranking of most cited papers and top author keywords, we can forecast that the environmental perspective will keep attracting the attention of researchers and both governmental and industrial communities will make efforts in policy making and commercial practice. Future research might incorporate theoretical knowledge of environmental disciplines into traditional supply chain management theories, as well as use a cross-disciplinary research paradigm to result in the SCM research being more practically useful.

(2) Supply chain management (SCM) driven by new technology. In the context of the Industry 4.0 era, industrial revolution undoubtedly drives the innovation of traditional supply chain management, as well as the emergence of new technologies, which will make supply chain management more intelligent. Much literature on smart supply chain management has emerged under Industry 4.0 in recent years. Innovations in technology of smart supply chain management research include block chain-driven, big data analysis-driven, and artificial intelligence (AI)-driven applications. Integration of SCM with block chain technology [105–108] and big data analytics [109–111] have been two rapidly developing areas of interest, as we described above in Figure 8 and Table 8. Research on SCM will also benefit from artificial intelligence (AI) applications [112–115] as we predict. The trend of author keyword ranking clearly revealed that "Industry 4.0", "blockchain", and "big data" had exploded in popularity in recent years, so we can confidently forecast that technology-driven supply chain management will remain a popular research topic in the future.

(3) Supply chain management (SCM) in the context of digital economy. Different from the technology-driven SCM in category 2, a great number of digital platform-based enterprises have emerged in the digital economy, such as e-commerce platforms, live streaming platforms, short video platforms, social platforms, sharing platforms, and so on [116–121]. The platform-based supply chain differs significantly from the traditional supply chain, which is a chain structure. Instead, the platform-based supply chain is a mesh structure with multilateral market features, and the chain linkage relationship between supply chain members is end-to-end. Its new qualities need rebuilding and cutting the supply chain research model, as well as conducting research using multilateral market theory.

(4) Digital and Intellectual SCM. With the progress of the digital revolution, artificial intelligence (AI) and other intelligent technologies have become a key part of the digital revolution. Enterprises use AI technology to improve customer experience, create new business models, and combine digital capabilities with AI technology strategies [122]. Digital and intelligent technologies lead to profound innovations from both the supply and the demand side. As customers become more demanding, enterprises make innovations by providing Digital-Service-Product Packages (DSPPs) with integrated, open and expansible functions. Intelligent manufacturing based on cloud computing, artificial intelligence, robotics, and other digital intelligence technologies is booming worldwide [123]. As a result, digitalization and intellectualization have been integrated into supply chain management to innovate traditional manufacturing as well as become a new interdisciplinary field.

This article identifies the characteristics and research trends of SCM quantitatively and qualitatively. However, there are still some limitations. As we mentioned in the introduction, the bibliometric analysis methodology has its own limitations, which represent one of this article's limitations. The indicators of bibliometric analysis lack the complexity and numerous dimensions of research production, such as the h-index. Despite the fact that the Web of Science covers a large number of publications, valuable publications would still be consequently omitted. The reason is that other databases, such as Scopus and Google Scholar, may also include relevant publications. Further work should focus on more comprehensive data collection, more accurate analysis of literature characteristics and research trends, and a more in-depth examination of the reasons for the analysis of results. Future efforts should focus on more comprehensive data collection from different databases so that the characteristics of the literature and research trends can be more accurately analyzed, as well as conducting more in-depth research.

**Author Contributions:** Conceptualization and methodology, H.F.; writing—original draft preparation, F.F.; writing—review and editing, F.F. and Q.H.; investigation and visualization, H.F. and Y.W.; supervision, H.F. and Y.W.; project administration, Q.H.; funding acquisition, Q.H. and H.F. All authors have read and agreed to the published version of the manuscript.

**Funding:** This research was funded by the Development Foundation Project of Shanghai University of Finance and Economics Zhejiang College, China (grant number 2020GR004), the General Project of Shanghai University of Finance and Economics Zhejiang College, China (grant number 2019YJYB02), and the Department of Education of Zhejiang Province, China (grant number Y201635361).

**Institutional Review Board Statement:** Not applicable.

**Informed Consent Statement:** Not applicable.

**Data Availability Statement:** Not applicable.

**Conflicts of Interest:** We declare that we do not have any commercial or associative interests that represent a conflict of interest in connection with this work.

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
