# Peer review of "Supply Chain Management: A Review and Bibliometric Analysis"

_processes, doi:10.3390/pr10091681_

Round 1

Reviewer 1 Report

Supply Chain Management: A Review and Bibliometric Analysis. This paper is very well written, has a good flow, and presents the information effectively. While I was originally going to suggest that the order of the introduction change a bit, I feel that it is good the way it is.

The topic is interesting, necessary, and timely as well. The application of the bibliometric analysis is also well done.

I also feel that this paper has the potential to make a contribution to the supply chain management literature. Thank you for an informative paper.

Author Response

Thank you for your kind comments and recognition of our manuscript. We deeply appreciate your careful review. We wish good health to you, your family, and your community.

Reviewer 2 Report

The paper provides an overview of current trends in supply chain management research which makes the paper a relevant contribution to pool of knowledge. The aim is set as both ambitious and reachable. Individual arguments are well presented. Conclusions are consistent with the evidence. I would recommend to further elaborate limitations of the study related to selected methodology. 

Author Response

Thank you for your careful review and the suggestion is precious to us. We have elaborated on the limitations of the study related to the bibliometric analysis methodology in the penultimate paragraph of Section 1 (Introduction). It is revised in the manuscript as “However, the limitations of this approach should also be noted. The h-index was created as a straightforward indicator of output and effect combined due to its accessibility and simplicity. Although it has been widely used, these metrics lack the complexity and numerous dimensions of research production and effect because they are too basic.” We have cited the reference [57] here and would like to know whether it is appropriate to add the content in this position, and if not, we will adjust it again. We also revised the last paragraph of Section 6 to show the limitation of the methodology is one of the limitations of our manuscript. “As we mentioned in the introduction, the bibliometric analysis methodology has its own limitations, which is one of this article’s limitations. The indicators of bibliometric analysis lack the complexity and numerous dimensions of research production, such as the h-index.”

Thank you again for your busy schedule to review our manuscript. Your careful review has helped to make our study more comprehensive. Best wishes to you!

Reviewer 3 Report

An interesting paper that attempts to observe the development of SCM starting from 2010. The conclusion opens new nuances related to environmental aspects, new technology, and digital technology.

However there are several things that need improvement. The first is related to the motivation or background of this research. The reason for conducting a literature review on SCM is not that the previous LR only contained the scope of the environment and sustainability. This is not true, the author needs to know that there has been research on SCM, only overall or what is the difference between it and the existing one.

The second is the issue of the definition and limitations of SCM itself, in this paper it is not explained what SCM is and its limitations that are used or become the output of this literature.

Third, it is not clear what areas emerged from the results of the analysis or findings, whether there are new interdisciplinary or interdisciplinary ones.

Fourth, it is not clear what the contribution of this research is, what can be proposed for further research, and what is new.

The author should be able to improve the four items mentioned above.
